# HyperVLA: Efficient Inference in Vision-Language-Action Models via Hypernetworks

## Abstract

Built upon language and vision foundation models with strong generalization ability and trained on large-scale robotic data, Vision-Language-Action (VLA) models have recently emerged as a promising approach to learning generalist robotic policies. However, a key drawback of existing VLAs is their extremely high inference costs. In this paper, we propose HyperVLA to address this problem. Unlike existing monolithic VLAs that activate the whole model during both training and inference, HyperVLA uses a novel hypernetwork (HN)-based architecture that activates only a small task-specific policy during inference, while still retaining the high model capacity needed to accommodate diverse multi-task behaviors during training. Successfully training an HN-based VLA is nontrivial so HyperVLA contains several key algorithm design features that improve its performance, including properly utilizing the prior knowledge from existing vision foundation models, HN normalization, and an action generation strategy. Compared to monolithic VLAs, HyperVLA achieves a similar or even higher success rate for both zero-shot generalization and few-shot adaptation, while significantly reducing inference costs. Compared to OpenVLA, a state-of-the-art VLA model, HyperVLA reduces the number of activated parameters at test time by $90\times$, and accelerates inference speed by $120\times$.

## 1 Introduction

Motivated by the great success of foundation models in domains like NLP (GLM et al., 2024; Jiang et al., 2023; Yang et al., 2025; Bai et al., 2023a; DeepSeek-AI et al., 2025; xAI, 2025; Team et al., 2025a; OpenAI et al., 2024; Grattafiori et al., 2024) and CV (Dosovitskiy et al., 2021; Radford et al., 2021a; Yu et al., 2022; Kirillov et al., 2023; Oquab et al., 2024; Wang et al., 2023; Bai et al., 2023b; Chen et al., 2025; Team et al., 2025b) in recent years, robotic learning has been going through a paradigm shift from training moderate-size models on a narrow task distribution to training generalist control policies on large-scale robotic demonstration data collected from a diverse set of real-world scenarios (Firoozi et al., 2023; Hu et al., 2023). Vision-Language-Action (VLA) models (Brohan et al., 2022; 2023; O'Neill et al., 2024; Team et al., 2024; Kim et al., 2024; Black et al., 2024) are one important family of such models, which take language instructions and image observations as input and predict the robot's action output. They usually use existing language and vision foundation models as the backbone to improve generalization, and are further trained on large-scale robotic data to learn the complex mapping from multi-modal inputs to the robot's action output.

While VLAs have shown promising generalization, one key drawback of these models is their extremely high inference costs, e.g., OpenVLA (Kim et al., 2024), a state-of-the-art (SOTA) VLA model, has more than 7B parameters and can only infer at 6Hz even when equipped with an NVIDIA 4090 GPU. Such a high inference cost not only consumes significant memory, computation, and energy, but also makes it hard to solve dexterous tasks that require high-frequency manipulation.

By contrast, conventional methods for robotic learning from before the era of foundation models typically learn compact models that are much smaller than VLAs. Although such models cannot generalize across a diverse set of tasks, they can perform well on the specific task they are trained on given sufficient training data. Hence, the minimal model required to solve a specific task can be much smaller than a VLA with millions or billions of parameters. So a natural question arises:

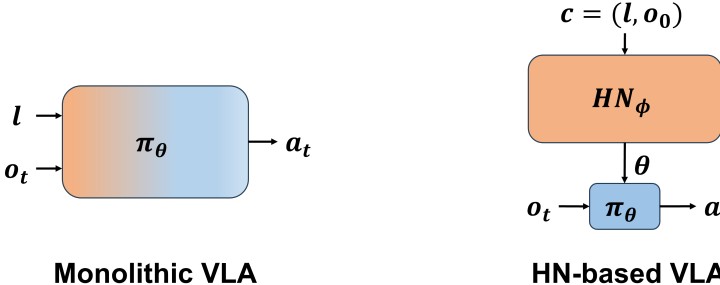

Figure 1: Comparison between the high-level framework of monolithic VLA (left) and HN-based VLA (right). We use orange to represent parameters activated during training, and blue to represent parameters activated at every timestep during inference. The monolithic VLA activates the whole model during both training and inference and is thus colored both orange and blue. By contrast, an HN-based VLA calls the HN at a low frequency only at the beginning of a new episode at test time, and calls a compact base network at every timestep for action prediction.

*Can we learn a generalist policy that combines the best of both worlds: the strong generalization ability of VLAs, and the efficient inference of single-task policies?* To achieve this, we need to learn a generalist policy with high model capacity to accommodate the diverse behaviors in multi-task data at training time, but only activate a small part of it at test time to keep inference efficient.

In this paper, we realize this goal via hypernetworks (HNs) (Ha et al., 2016). An HN is a network that generates the parameters of another base network conditioned on some context information. Its hierarchical architecture provides a natural way to decouple the skills required to solve different tasks (Xiong et al., 2024), so that we can learn an HN with high model capacity at training time, but only activate a compact HN-generated base network to solve a specific task efficiently at test time.

Therefore, we learn an HN-based VLA that generates policy parameters conditioned on task context $c$, which in our setting consists of both the language instruction $l$ and the initial image $o_0$ of an episode (Figure 1). At training time, we train an HN with high model capacity to capture the complex mapping from the task context $c$ to the corresponding policy parameters $\pi_\theta^c$. At inference time, the large HN is called at a low frequency, only when the task context changes at the beginning of a new episode, while the compact generated policy is called at every timestep to process image observations and output action predictions, which significantly reduces inference cost compared to existing monolithic VLAs that activate the entire model at every timestep during inference.

However, HNs are known to be hard to optimize (Chang et al., 2020; Beck et al., 2023a; Xiong et al., 2024), and training an HN with millions of parameters on large-scale robotic data further aggravates this issue. We thus introduce several key algorithm design features to improve HN learning:

1. **Vision backbone:** While in principle we can generate the whole base policy with an HN, empirically we find it important to use existing vision foundation models as the backbone to improve generalization, as training an HN from scratch on existing robotic datasets, which are relatively small, is prone to overfitting.

2. **HN normalization:** Successful training of neural networks depends heavily on many optimization design choices, which are mainly tailored for training monolithic models and may not generalize to the different optimization dynamics of HNs (Chang et al., 2020; Beck et al., 2023a; Auddy et al., 2024). We thus investigate how parameter updates in HN training differ from standard training, and propose a simple yet effective solution by normalizing the context embedding in HNs, such that the base network parameters can be updated with similar dynamics as directly training the base network.

3. **Action generation strategy:** Unlike most existing VLAs that predict actions via autoregression (Brohan et al., 2022; 2023) or diffusion (Chi et al., 2023; Team et al., 2024), we find that learning a simple linear action head with MSE loss performs better when training an HN-based VLA and further accelerates inference.

We call our method HyperVLA, an HN-based VLA learned with the above algorithm design features. We train HyperVLA on the Open X-Embodiment (OXE) dataset (O'Neill et al., 2024), and evaluate it for both zero-shot generalization to seen and unseen tasks from the training scenarios, and few-shot adaptation to new domains. Compared to existing monolithic VLAs, HyperVLA achieves a similar or higher success rate during evaluation, while significantly improving inference efficiency by only activating a compact HN-generated policy at every timestep during inference, which validates the effectiveness of utilizing HNs for inference acceleration of VLAs. Compared to OpenVLA, a SOTA VLA with the best performance among the baselines, HyperVLA reduces the number of activated parameters at test time by $90\times$, and accelerates inference by $120\times$.

## 2 BACKGROUND

### 2.1 VISION-LANGUAGE-ACTION MODELS

Vision-Language-Action (VLA) models aim to learn a generalist robotic control policy, which takes in a language instruction $l$ and image observations $o_t$ at each timestep $t$ and predicts the robot's action $a_t$. In this paper, we focus on image observations, though other input modalities can be easily integrated into our approach. To achieve good generalization, VLAs are usually built upon vision and language foundation models and trained on large-scale robotic data via behavior cloning (BC). The robotic dataset consists of expert demonstrations collected for different tasks in different scenarios. Each demonstration episode consists of a sequence of image observations and corresponding actions $(o_0, a_0, \ldots, o_{T-1}, a_{T-1}, o_T)$, and optionally a language instruction $l$ if annotated. For each expert observation-action pair $(o_t, a_t)$, the BC loss is defined as $L_{\text{BC}} = (\hat{a}_t - a_t)^2$, where $\hat{a}_t = \pi_\theta(o_t, l)$ is the action predicted by the policy.

### 2.2 HYPERNETWORKS

A hypernetwork (Ha et al., 2016) is a network that generates some or all of the parameters of a base network conditioned on some context $c$. This hierarchical architecture offers a powerful tool for multi-task robotic control, as we can generate different policies for different tasks conditioned on their task context. The parameters $\theta$ of the base network can be divided into $\theta^{\text{generated}}$ generated by the HN, and $\theta^{\text{shared}}$ which is not generated by the HN and shared across all the tasks. To generate $\theta^{\text{generated}}$, the HN first encodes the task context with a context encoder $f$ to get a context embedding $e^{\text{context}} = f(c)$, then passes the context embedding through linear output heads to predict $\theta^{\text{generated}}$.

## 3 HYPERVLA

This section introduces the motivation, architecture, and algorithm design of HyperVLA. In Section 3.1, we analyze why existing monolithic VLAs have high inference costs and how an HN-based VLA can tackle this challenge. Then in Section 3.2 we introduce the architecture of HyperVLA. Finally, in Section 3.3, we propose several key algorithm design features to stabilize HyperVLA training and improve its performance.

### 3.1 FROM MONOLITHIC TO HN-BASED VLA

Existing VLAs usually have millions or billions of parameters. While such a high model capacity is necessary to accommodate diverse behaviors in multi-task data at training time, it introduces significant computational redundancy at test time as the minimal model required to solve a specific task is often much smaller than a large VLA (Yu et al., 2020; Kim et al., 2024).

Consequently, there is room for inference acceleration if we can only activate a small part of a huge VLA that is sufficient to solve the task at hand. However, since existing VLAs have monolithic architectures that require activating the whole model during both training and inference, they cannot decouple in parameter space the different skills required to solve different tasks (Xiong et al., 2024).

In this paper, we tackle this challenge by learning an HN-based VLA, as the hierarchical architecture of HNs provide a natural way to decouple inter-task and intra-task knowledge. Intuitively, the HN is a generalist that encodes inter-task knowledge about how to map from different task context to the

corresponding policy parameters, while the base network generated by the HN is a specialist that encodes intra-task knowledge about how to solve a specific task. At training time, the whole HN is activated to ensure sufficient model capacity to accommodate the diverse behaviors in multi-task data. However, at test time, we only need to call the HN once at the beginning of each episode to generate a compact task-specific policy that is used for the remainder of the episode.

### 3.2 THE ARCHITECTURE OF HYPERVLA

**The base policy**   We formulate the base policy as a Vision Transformer (ViT) (Dosovitskiy et al., 2020), which takes the image observation $o_t$ as input to predict the robot's action $a_t$. Unlike existing VLAs, we do not feed the language instruction $l$ into the base policy as it is already indirectly conditioned on the instruction via the HN that generated its parameters. The base policy consists of the following blocks in sequence (we omit the time index $t$ for simplicity):

1. An image encoder, formulated as a ViT, encodes the image observation $o$ into a sequence of token embeddings $\{e_i^{\text{image}}\}$ for the image patches;

2. A linear projection layer maps $\{e_i^{\text{image}}\}$ into a lower dimension for more efficient inference, represented as $\{e_i^{\text{proj}}\}$;

3. A policy head $\theta^{\text{policy}}$, formulated as a small Transformer, takes $\{e_i^{\text{proj}}\}$ and a learnable action token $e^{\text{act}}$ as inputs, and updates their token embeddings; and

4. An action head takes updated $e^{\text{act}}$ as input to predict the robot's action $\hat{a}$.

**The hypernetwork**   The HN consists of a context encoder parameterized as a Transformer with high model capacity, and linear output heads to generate base policy parameters. The context encoder takes in three inputs:

1. Pretrained instruction embeddings generated by a frozen T5 encoder (Raffel et al., 2020);

2. The class token embedding of the initial image generated by a frozen DINOv2 encoder. We find it helpful to condition the HN on the initial image, as the robot may see the same instruction in different scenarios, and a compact base policy may not have enough model capacity to solve the same task across scenarios with diverse visual appearance. By conditioning policy generation further on the initial image, the HN with high model capacity takes the responsibility of generalization across both instructions and scenarios, while the generated base policy only needs to solve a specific task in a specific scenario. Moreover, we only use the class token outputted by DINOv2 as HN input and discard the image patch tokens to avoid overfitting in the HN; and

3. A learnable task context token that integrates task context information.

The context encoder updates the embeddings of these tokens via self-attention, and the embedding of the task context token is fed into the HN output heads to generate base policy parameters.

### 3.3 ALGORITHM DESIGN FEATURES

HNs are known to be unstable and hard to optimize (Chang et al., 2020; Beck et al., 2023a; Xiong et al., 2024), and scaling them up to millions of parameters further aggravates this issue. We thus introduce several key algorithm design features that help stabilize and improve HyperVLA training.

#### 3.3.1 VISION BACKBONE

While in principle we can generate the whole base policy by HN, empirically we find that training a large HN from scratch on robotic data alone is prone to overfitting due to the relatively small data size of existing robotic datasets. Instead, we use existing vision foundation models as the image encoder in the base network to improve generalization. Our method is agnostic to the choice of the vision encoder, and empirically we find that DINOv2 (Oquab et al., 2024) achieves the best performance and thus adopt it in HyperVLA.

Similar to previous work (Kim et al., 2024), we find it helpful to fine-tune this vision backbone instead of keeping it frozen when training on robotic data. Furthermore, we use a smaller learning rate for fine-tuning the vision backbone than for HN training because DINOv2 is already well pre-trained and only needs to be fine-tuned at a conservative rate to better align with robotic data.

### 3.3.2 Context embedding normalization

Successful training of neural networks depends heavily on many optimization choices, such as network initialization, normalization layers, and gradient transformations. However, such choices are mainly tailored to monolithic models, and may need to be redesigned to fit the different optimization dynamics of HNs. For example, Chang et al. (2020) and Beck et al. (2023a) investigate how to initialize HNs properly so that the base network is initialized in the same way as commonly used initializers, an approach we adopt as well.

However, a proper initialization can only provide a good starting point for HN training and has no direct effect on the parameter update process during training. So in this paper, we further investigate how parameter updates in HN training differ from standard neural network training, and propose a simple yet effective solution by normalizing the context embedding in HNs, such that the base network parameters can be updated with similar dynamics as directly training the base network.

For simplicity, we use SGD in our derivation below. In standard neural network training, each parameter $\theta_i$ is updated by $\Delta\theta_i = -\alpha \cdot \frac{\partial L}{\partial \theta_i}$, where $L$ is the loss function and $\alpha$ is the learning rate.

Now we generate $\theta$ with an HN. Let us denote the output head of the HN as $\phi$, and the context embedding input to the output head as $e$, then we have $\theta_i = \sum_j e_j \phi_{ij}$. We omit the bias term as it has the same gradient as the base parameter. According to the chain rule, we have $\frac{\partial L}{\partial \phi_{ij}} = \frac{\partial L}{\partial \theta_i}\frac{\partial \theta_i}{\partial \phi_{ij}} = \frac{\partial L}{\partial \theta_i}e_j$, and the parameter update in the HN is $\Delta\phi_{ij} = -\alpha \cdot \frac{\partial L}{\partial \theta_i}e_j$. Then the parameter update in the base network is:

$$\Delta\theta_i = \sum_j (e_j + \Delta e_j)(\phi_{ij} + \Delta\phi_{ij}) - \sum_j e_j\phi_{ij}, \tag{1}$$

$$= \sum_j e_j\Delta\phi_{ij} + \Delta e_j\phi_{ij} + \Delta e_j\Delta\phi_{ij}, \tag{2}$$

$$\approx \sum_j e_j\Delta\phi_{ij} + \Delta e_j\phi_{ij}, \quad \text{(Omit the multiplication of two delta terms)} \tag{3}$$

$$= -\alpha \cdot \frac{\partial L}{\partial \theta_i}\sum_j e_j^2 + \sum_j \Delta e_j\phi_{ij}. \tag{4}$$

If we assume that both $\phi_{ij}$ and $\Delta e_j$ are i.i.d. and follow a Gaussian distribution with zero mean, then $\mathbb{E}\left[\sum_j \Delta e_j\phi_{ij}\right] = 0$. Accordingly, we have $\Delta\theta_i \approx -\alpha \cdot \left(\sum_j e_j^2\right) \cdot \frac{\partial L}{\partial \theta_i}$, which indicates that when learning with HNs, the update on the base network parameters is scaled by a factor of $\sum_j e_j^2$ compared to directly optimizing the base network.

In HyperVLA, as the context embedding $e$ is the output of a Transformer context encoder with layer normalization as its final layer, we have $\mathbb{E}\left[e_j^2\right] = 1$ and $\mathbb{E}\left[\sum_j e_j^2\right] = d_e$, where $d_e$ is the dimension of $e$. Consequently, to keep the scale of parameter update in the base network unchanged, we can simply divide the context embedding by $\sqrt{d_e}$ before feeding it into the output head so that $\mathbb{E}\left[\sum_j e_j^2\right] = 1$ after normalization.

The derivation is different for more complex optimizers like Adam, which makes it much harder to theoretically keep the update scale unchanged like with the SGD optimizer. However, empirically we find that the same normalization operation of dividing the context embedding by $\sqrt{d_e}$ before feeding it into the HN output head still works well in practice.

### 3.3.3 Action generation strategy

Existing VLAs usually predict discretized actions autoregressively (Brohan et al., 2022; 2023; Kim et al., 2024), which requires multiple runs of the same model to generate different action dimensions

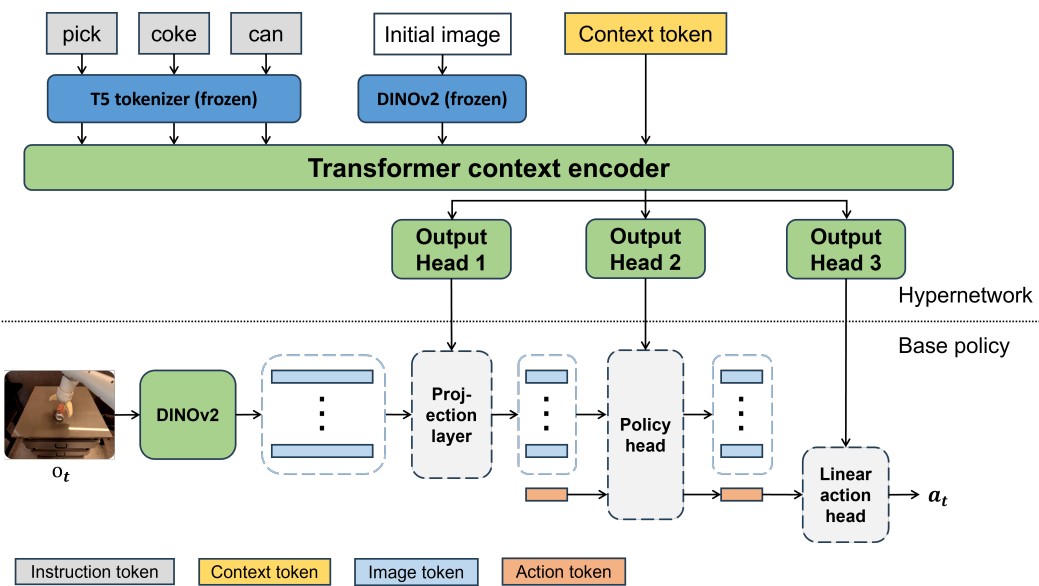

Figure 2: The framework of HyperVLA. The trainable parameters are marked as green blocks, while the HN-generated parameters are marked as light grey blocks with dashed edges.

sequentially, or learn a diffusion action head (Chi et al., 2023) that must be iteratively called to denoise actions (Team et al., 2024), both of which are time-consuming at training and test time. Instead, we find that training a simple linear action head with an MSE loss outperforms these more complicated action generation strategies in HyperVLA, while further reducing training and inference cost. This also agrees with the findings from some recent work (Kim et al., 2025).

Combining these features, the overall framework of HyperVLA is shown in Figure 2.

## 4 EXPERIMENTS

Our experiments aim to answer the following questions:

- **Q1:** Can HyperVLA match the zero-shot generalization performance of existing monolithic VLAs on both seen and unseen tasks? (Section 4.2)
- **Q2:** Can HyperVLA adapt to new tasks by fine-tuning on only a few demonstrations, especially for long-horizon tasks? (Section 4.3)
- **Q3:** Can HyperVLA be more inference efficient than monolithic VLAs? (Section 4.4)
- **Q4:** How do the algorithm designs in HyperVLA influence its performance? (Section 4.5)

### 4.1 EXPERIMENTAL SETUP

**Baselines** We compare HyperVLA with the following monolithic VLAs as baselines: (1) **RT-1-X** (O'Neill et al., 2024): uses EfficientNet (Tan & Le, 2019) as the vision backbone, and conditions on the language instruction via FiLM (Perez et al., 2018). Each action dimension is discretized and predicted autoregressively. It has roughly 35M parameters. (2) **Octo** (Team et al., 2024): uses T5 (Raffel et al., 2020) as the language backbone and learns the visual encoder on robotic data alone. It predicts actions via policy diffusion (Chi et al., 2023). It has roughly 200M parameters. (3) **OpenVLA** (O'Neill et al., 2024): uses SigLIP (Zhai et al., 2023) and DINOv2 (Oquab et al., 2023) as the vision backbones, and Llama 2 (Touvron et al., 2023) as the language backbone. Llama 2 is further fine-tuned on robotic data to predict action tokens autoregressively like RT-1-X. It has about 7.6B parameters. All the baselines are trained on the Open X-Embodiment (OXE) dataset (O'Neill et al., 2024), which contains demonstrations collected from different robot embodiments.

We choose these models as the baselines as they are all trained on the OXE dataset and have the same input and output space for the model, which makes it easier to control variates and focus only on the influence of changing the model architecture from monolithic to HN-based. Many later VLAs are trained on larger and different datasets, which makes it hard to tell whether the performance difference is caused by the data or the use of HN. Nevertheless, future work can easily apply our HN-based architecture to other VLAs by adopting their original training recipe for a fair comparison.

**Hyperparameters of HyperVLA** In the base network, we use DINOv2 (Oquab et al., 2023) as the image encoder. The policy head is a Transformer with 4 layers, each with 4 attention heads. Its token embedding dimension and hidden layer dimension are set to 64 and 128 respectively. The base network takes only the current image observation as input, and predicts an action chunk of 4 steps. During evaluation, we further apply action ensemble (Zhao et al., 2023), i.e., averaging the last 4 steps' action predictions on the current step, to improve prediction accuracy. For the HN, we adopt T5 (Raffel et al., 2020) as the instruction encoder and DINOv2 as the image encoder, and freeze them during training. We learn a Transformer context encoder with 6 layers, with an embedding dimension of 128, MLP hidden dimension of 512 and 4 attention heads for each layer. The output heads are linear layers that map the context embedding to the base parameters.

**Training setup** For a fair comparison with the baselines, we also train HyperVLA on the OXE dataset. We train it for 100k steps with a batch size 256. See Appendix B.1 for the detailed training setup. We include the source code in the supplementary material to facilitate reproducibility.

## 4.2 Zero-shot generalization results

| Robot | Google Robot | | | | WidowX | | | | |
|-------|------|------|-------------|------|----------------|-----------------|------------|--------------------|-------|
| Task set | pick | move | close drawer | Avg. | spoon on towel | carrot on plate | stack cube | eggplant in basket | Avg. |
| ID or OOD | both | ID | ID | | ID | ID | OOD | OOD | |
| RT-1-X | 29 | 46 | **65** | 47 | 10 | 12 | 0 | 0 | 6 |
| Octo | 7 | 26 | 31 | 21 | 5 | 1 | 0 | 45 | 13 |
| OpenVLA | 10 | **72** | 54 | 45 | 25 | **18** | 34 | 65 | 36 |
| **HyperVLA** | **58 ± 3** | **73 ± 1** | 58 ± 7 | **63 ± 3** | **48 ± 3** | 21 ± 5 | **39 ± 8** | 52 ± 13 | **40 ± 5** |

Table 1: Evaluation success rates of different methods on SIMPLER. For Google Robot, each column represents a task set which contains multiple different instructions, and we report the average success rate over the whole task set. The "ID or OOD" row represents if the task set is in-distribution (ID) or out-of-distribution (OOD). For our method, we report the performance mean and standard error averaged over 5 random seeds. For the baselines, we cannot report the confidence interval, as only a single model checkpoint is publicly available for each baseline method.

To answer Q1 about zero-shot generalization performance, We evaluate on the SIMPLER benchmark (Li et al., 2024b), which reproduces some tasks from the OXE dataset in simulation and is specifically designed to align with real-world evaluation results, so that different VLAs can be compared in a reproducible way. SIMPLER includes two commonly used robot arms, Google Robot and WidowX, and defines a set of different tasks for each robot. SIMPLER evaluates on both tasks that have been seen during training with different demonstration number ranging from 1 to more than 2,000, and unseen tasks with new instructions (see Appendix B.2 for more details). For seen tasks, generalization across parametric variations is evaluated, such as object layout, position and orientation. For unseen tasks, generalization across instructions is further evaluated.

Table 1 compares the success rates of different methods on the SIMPLER benchmark. Among the baselines, OpenVLA performs the best overall, as it builds upon strong language and vision foundation models which facilitate generalization, but at the expense of a higher inference cost. Our method achieves similar performance to OpenVLA on most task sets, while significantly outperforming all baselines on the picking task set. This validates that HyperVLA can significantly reduce training and inference cost (Section 4.4) without sacrificing performance.

## 4.3 Few-shot adaptation results

To answer Q2 about few-shot adaptation performance, we evaluate on the LIBERO benchmark (Liu et al., 2023), which is commonly used to evaluate data-efficient fine-tuning of VLAs. Following

the same setup as in OpenVLA, we evaluate on four task suites in LIBERO, i.e., LIBERO-Spatial, LIBERO-Object, LIBERO-Goal and LIBERO-Long, each containing 10 tasks (instructions) with 50 demonstrations for each task (see Appendix B.2 for more details). For a fair comparison, we preprocess the demonstrations in the same way as OpenVLA. We fine-tune HyperVLA on the first three task suites for 10k steps, and LIBERO-Long for 60k steps as it constitutes of long-horizon tasks that are harder to solve. All the other hyperparameters are set in the same way as for pretraining.

As shown in Table 2, our method significantly outperforms Octo and OpenVLA on all the task suites after fine-tuning, validating the effectiveness of our method for few-shot adaptation to unseen tasks. The significant advantage of HyperVLA on LIBERO-Long further validates that it can also solve complicated long-horizon tasks by only activating a compact base policy at inference time.

|  | LIBERO-Spatial | LIBERO-Object | LIBERO-Goal | LIBERO-Long | Average |
|---|---|---|---|---|---|
| Octo | 79 | 86 | 85 | 51 | 75 |
| OpenVLA | 85 | 88 | 79 | 54 | 77 |
| **HyperVLA** | **95** | **94** | **92** | **74** | **89** |

Table 2: Evaluation success rate of different methods on LIBERO after fine-tuning. We evaluate on each task for 50 episodes. The results of the baselines are taken from Kim et al. (2024).

### 4.4 INFERENCE EFFICIENCY

To answer Q3 about inference efficiency, Table 3 compares the number of parameters activated during training and inference, the inference speed, and FLOPs of different methods. For the activated parameters at inference time, we exclude the instruction encoder in all the methods and the HN in our method, as they are only activated once at the beginning of each episode, and their computational costs are negligible compared to the total inference cost across the whole episode.

| Method | # params activated for training | # params activated for test | Time per inference step (ms) | FLOPs |
|---|---|---|---|---|
| RT-1-X | 35M | 35M | 88 | - |
| Octo | 200M | 100M | 96 | $5.6 \times 10^{10}$ |
| OpenVLA | 7.6B | 7.6B | 482 | $4.0 \times 10^{12}$ |
| **HyperVLA** | 86M (shared) + 216M (HN) | 86M (shared) + 0.1M (generated) | **4** | $\mathbf{4.7 \times 10^{10}}$ |

Table 3: Number of parameters activated during training and test, inference speed, and FLOPs of different methods. Time per inference step is measured by running each model on an NVIDIA L4 GPU. We were unable to measure the FLOPs of RT-1-X as its model checkpoint is wrapped up.

While HyperVLA learns both a shared DINOv2 image encoder with 86M parameters and an HN with 216M parameters (100M for the frozen T5 encoder + 86M for the frozen DINOv2 encoder + 30M for the learned context encoder) during training, at test time it only activates the shared DINOv2 backbone and a compact base network with 0.1M parameters for each inference step, which leads to a significant acceleration. Compared to OpenVLA, the baseline with the best performance, HyperVLA reduces the model size at inference time 90-fold and accelerates the inference speed 120-fold. Although RT-1-X and Octo have a similar or smaller number of activated parameters during inference than HyperVLA, their inference is still much slower, as they use either autoregression or a diffusion policy to predict the action, both of which require more iterations over the model parameters than the simple linear action head used in HyperVLA.

Based on the above results, we conclude that HyperVLA not only achieves similar or better performance compared to the baselines, but also significantly reduces inference costs. Moreover, while our primary goal is to improve the inference efficiency of VLAs, our method also significantly reduces computational costs during training. Specifically, OpenVLA is trained on 64 A100 GPUs for 14 days (Kim et al., 2024), while HyperVLA can be trained on just 4 A5000 GPUs in a single day.

### 4.5 ABLATION STUDIES

To answer Q4 about the effectiveness of the algorithm designs in HyperVLA, we run ablation studies by removing each of the algorithm design features proposed in Section 3.3 from it. The ablation results in Table 4 validate that all the proposed designs contribute to the success of HyperVLA.

**Vision backbone:** When removing the DINOv2 backbone, we increase the number of training steps to 600k for a fair comparison, as training the whole model from scratch takes longer to converge. However, even with a larger training budget, it still significantly underperforms HyperVLA, illustrating the importance of utilizing the prior knowledge from vision foundation models.

**HN normalization:** To ablate HN normalization, we do not normalize the context embedding before feeding it into the HN output heads. In general, this variant performs slightly worse than HyperVLA on seen tasks, but significantly worse on the two OOD WidowX tasks, which validates the importance of stabilizing HN learning with context embedding normalization.

**Action generation strategy:** We replace the linear action head in HyperVLA with a diffusion action head like in Octo (Team et al., 2024), and increase the training steps to 400k. The diffusion-head variant underperforms HyperVLA, which illustrates that a simple linear action head trained with MSE loss is sufficient when training an HN-based VLA, and also improves training efficiency compared to more complicated action head designs like diffusion.

Please see Appendix C.1 for ablation results on more detailed design choices in HyperVLA.

| Method | Google robot | | | | WidowX | | | | |
|---|---|---|---|---|---|---|---|---|---|
| | pick | move | close drawer | Avg | spoon on towel | carrot on plate | stack cube | eggplant in basket | Avg |
| HyperVLA (Full) | $58 \pm 3$ | $73 \pm 1$ | $58 \pm 7$ | $63 \pm 3$ | $48 \pm 3$ | $21 \pm 5$ | $39 \pm 8$ | $52 \pm 13$ | $40 \pm 5$ |
| - Vision backbone | 24 | 30 | 38 | 31 | 21 | 0 | 0 | 0 | 5 |
| - HN normalization | $53 \pm 4$ | $71 \pm 4$ | $48 \pm 1$ | $57 \pm 1$ | $48 \pm 3$ | $27 \pm 6$ | $19 \pm 3$ | $31 \pm 9$ | $31 \pm 4$ |
| - Linear action head | 49 | 56 | 55 | 53 | 42 | 23 | 15 | 39 | 30 |

Table 4: Ablations on how different algorithm designs in HyperVLA influence its performance.

## 5 RELATED WORK

Using language and vision foundation models as backbone enables VLAs to generalize across a broad range of tasks at the expense of high inference cost. Using smaller backbone models is thus a straightforward way to accelerate VLAs (Belkhale & Sadigh, 2024; Wen et al., 2025; Shukor et al., 2025). Predicting action chunks (Zhao et al., 2023; Team et al., 2024; Black et al., 2024) instead of a single-step action is another common approach, so the VLA does not need to be called at every timestep. The idea of learning a large VLA but only partially activating it during inference has also been explored: DeeR-VLA (Yue et al., 2024) early exits from intermediate layers if the layer output is sufficient for action prediction. Closely related to the hierarchical architecture in our method, dual-system VLAs (Shentu et al., 2024; Han et al., 2024; Zhang et al., 2024; Bu et al., 2024; Cui et al., 2025) learn both a high-level planner that generates a latent goal and operates at a low frequency, and a low-level policy that conditions on this latent goal to generate per-step actions. Compared to existing methods, HyperVLA accelerates VLA inference in an orthogonal way by decoupling the skills required to solve different tasks via HNs and can be combined with existing approaches for further acceleration, such as parameterizing the high-level and low-level models in dual-system VLAs as HNs. Due to space limitation, see Appendix A for more related work on general VLAs and context-conditioned policy generation beyond the domain of VLAs.

## 6 CONCLUSION

In this paper, we analyzed why existing VLAs have high inference cost due to their monolithic architectures, and proposed an HN-based solution that decouples the skills required to solve different tasks at test time for inference acceleration. To stabilize HN training and improve its performance, we further proposed several key algorithm design features, including how to properly integrate vision backbones, HN normalization, and a simple linear action head trained with MSE loss. Building upon HN's ability to decouple the skills to solve different tasks and this algorithm design, we proposed HyperVLA, which achieves performance similar to or even better than that of existing monolithic VLAs, while significantly improving inference efficiency.

Our paper opens many interesting directions for future work on HN-based VLAs, such as evaluating on real robots, scaling up the HN model size, and training on more recent and larger robotic datasets (Khazatsky et al., 2024; Bjorck et al., 2025) for further performance improvement, and integration with task planning to solve more complicated long-horizon tasks.

REPRODUCIBILITY STATEMENT

We have made the following efforts to ensure the reproducibility of our work:

1. We clearly describe our method in Section 3, including both the detailed architecture of HyperVLA in Section 3.2, and the key algorithm designs in Section 3.3.

2. We include the source code to reproduce both HyperVLA and the baseline results in the supplementary material.

3. We use publicly available datasets and benchmarks (OXE, SIMPLER, and LIBERO) for experiments, and follow the same data preprocessing pipeline as in previous work (Team et al., 2024; Kim et al., 2024).

4. We clearly describe the hyperparameters of experiments in Section 4.1 and Appendix B.1 to facilitate reproducibility.

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

## A FURTHER RELATED WORK

**VLAs** Inspired by the success of foundation models in NLP and CV, RT-1 (Brohan et al., 2022) is a pioneering VLA work that validates the effectiveness of pretraining foundation models on large-scale robotic data. RT-2 (Brohan et al., 2023) further builds VLAs upon existing language and vision backbones to utilize their strong generalization ability, instead of learning a generalist controller from scratch on robotic data alone. O'Neill et al. (2024) propose the OXE dataset which validates the effectiveness of learning from cross-embodiment robotic datasets. Built upon these key ideas, more recent work investigates the design choices in VLAs in more detail, yielding further improvement from scaling up training data and learning from unlabeled videos (Black et al., 2024; Bu et al., 2025; Bjorck et al., 2025), the choice of backbone models (Kim et al., 2024; Li et al., 2024a), action representation and generation strategy (Team et al., 2024; Black et al., 2024; Song et al., 2025), fine-tuning on downstream tasks (Kim et al., 2024; 2025), etc. Please see Ma et al. (2024); Din et al. (2025); Kawaharazuka et al. (2025); Zhong et al. (2025) for more detailed reviews on VLAs.

**Context-conditioned policy generation** Faccio et al. (2023) and Di Ventura et al. (2025) adopt HNs to generate policies that can achieve different amounts of expected return in a single environment, and achieve promising performance on relatively simple continuous control tasks. Generating task-conditioned policies via HNs has been investigated in multi-task and meta-RL (Yu et al., 2019; Sarafian et al., 2021; Beck et al., 2022; 2023b; Rezaei-Shoshtari et al., 2023), but such work mainly focuses on learning lightweight models on narrow task distributions, and do not use HNs for inference acceleration. Make-An-Agent (Liang et al., 2024) treats parameter generation as a denoising process in the parameter space, and generates policy parameters via diffusion conditioned on demonstration trajectories, while our method generates the policy via HNs conditioned on the task context and can thus zero-shot generalize to a new task without additional demonstration trajectories from the new task. Closely related to our work, HyperDistill (Xiong et al., 2024) uses HNs to generate compact locomotion policies for efficient inference on different robot embodiments. Our method shares a similar motivation of accelerating inference via HNs but investigates a much more challenging setting of language-conditioned control with image observations and tackles the instability and generalization challenges in HN training at a much larger model scale.

## B ADDITIONAL EXPERIMENTAL SETUP

### B.1 HYPERVLA TRAINING SETUP

To stabilize the performance of the learned model, we apply exponential moving average to the model parameters with a smoothing factor of 0.999, and save the smoothed parameters instead of the latest parameters in the model checkpoints for evaluation. We optimize with AdamW (Loshchilov & Hutter, 2017), with a weight decay coefficient of 0.05 on HN output heads. Following the setup in Octo (Team et al., 2024), we set the peak learning rate as 3e-4, and apply learning rate warmup for 2k steps, then anneal it with an inverse square root schedule. The learning rate for the DINOv2 image encoder in the base network shares the same schedule, but uses a much lower peak value of 3e-5. To enable better generalization, we augment both the language instruction by rephrasing, and the image observation by image augmentation as done in Octo. Other training hyperparameters follow the same setup as in Octo.

### B.2 EVALUATION BENCHMARKS

**SIMPLER** Table 5 shows more detailed information about the evaluation tasks included in the SIMPLER benchmark.

**LIBERO** We evaluate few-shot adaptation of HyperVLA on the same four task suites as used in OpenVLA:

1. **LIBERO-Spatial** evaluates generalization to different layouts of the same set of objects;

2. **LIBERO-Object** evaluates generalization to different object types with the same scene layout;

| Robot | Task suite | # Instruction | Seen during training | # Demonstration in OXE | # Eval |
|---|---|---|---|---|---|
| Google Robot | pick | 13 | 6 seen, 7 unseen | ~600 per seen object | 150 |
| | move | 30 | Yes | 40 to 100 per instruction | 180 |
| | close drawer | 3 | Yes | > 2000 per instruction | 180 |
| WidowX | spoon on towel | 1 | Yes | 1 | 60 |
| | carrot on plate | 1 | Yes | 332 | 60 |
| | stack cube | 1 | No | 0 | 60 |
| | eggplant in basket | 1 | No | 0 | 60 |

Table 5: Summary of evaluation tasks in SIMPLER.

3. **LIBERO-Goal** evaluates generalization to different goals (instructions) with the same set of objects and layout; and

4. **LIBERO-Long** evaluates performance on long-horizon tasks with diverse objects, layouts, and tasks.

# C    ADDITIONAL EXPERIMENTAL RESULTS

## C.1    FURTHER ABLATION STUDIES

| Method | Google robot | | | | WidowX | | | | |
|---|---|---|---|---|---|---|---|---|---|
| | pick | move | close drawer | Avg | spoon on towel | carrot on plate | stack cube | eggplant in basket | Avg |
| HyperVLA (Full) | $58 \pm 3$ | $73 \pm 1$ | $58 \pm 7$ | $63 \pm 3$ | $48 \pm 3$ | $21 \pm 5$ | $39 \pm 8$ | $52 \pm 13$ | $40 \pm 5$ |
| Larger learning rate for DINOv2 | 3 | 13 | 17 | 11 | 0 | 0 | 0 | 0 | 0 |
| Frozen DINOv2 | 56 | 47 | 58 | 54 | 18 | 40 | 0 | 3 | 15 |
| Fine-tuned CLIP | 58 | 61 | 59 | 59 | 63 | 30 | 13 | 0 | 27 |
| Frozen SigLIP | 20 | 34 | 49 | 34 | 3 | 0 | 0 | 0 | 1 |
| Train base net alone | 5 | 11 | 12 | 9 | 3 | 0 | 0 | 0 | 1 |

Table 6: Ablation results on how different algorithm designs in HyperVLA influence its performance.

We report further ablation results in Table 6 to validate the importance of the following design choices in HyperVLA. To reduce computational cost, we run each ablation experiment with only one seed, while the performance gap is significant enough to draw conclusions with high confidence.

**Smaller learning rate for DINOv2 fine-tuning**    To ablate the importance of fine-tuning DINOv2 with a smaller learning rate as introduced in Section 3.3, we increase the learning rate for DINOv2 by 10 times, and use the same learning rate of 0.0003 for both HN training and DINOv2 fine-tuning. This variant performs poorly, which validates the importance of fine-tuning DINOv2 with a smaller learning rate to maintain its strong prior knowledge.

**Fine-tuning versus freezing DINOv2**    To validate the importance of fine-tuning DINOv2 in the base network, we ablate by freezing it while keeping the remaining settings unchanged. This variant underperforms HyperVLA, which validates the importance of fine-tuning DINOv2 during Hyper-VLA pretraining.

**Choice of the image encoder**    As introduced in Section 3.3, HyperVLA can support different image encoders, and empirically we find DINOv2 to perform best. We ablate by using either CLIP (Radford et al., 2021b) or SigLIP (Zhai et al., 2023) as the image encoder in the base network. As our code is implemented in JAX and we can only find a PyTorch version of SigLIP, our code does not support fine-tuning SigLIP during training. The ablation results show that using fine-tuned DINOv2 outperforms fine-tuned CLIP, while frozen DINOv2 outperforms frozen SigLIP.

**Importance of the HN**    We run this ablation experiment to validate that inference acceleration can not be achieved by training a small base network alone, and the HN in our method is essential to maintain high model capacity and achieve good performance. We experiment by removing the HN in HyperVLA and train the base network alone. This variant performs poorly, validating the importance of using HN.

## C.2 QUALITATIVE ANALYSIS

We include example videos of rolling out different methods on different tasks in the supplemental material, and qualitatively analyze the common failure patterns of different methods as follows:

The main failure reason of our method is inaccurate grasping of the object to manipulate, e.g., the policy sometimes may close the gripper when the end-effector is still slightly above the target object, which makes the robot fail to pick up the object. In general, the action error of HyperVLA is small and may be mitigated by integrating more camera views or further fine-tuning.

The OpenVLA baseline significantly underperforms our method on the picking task of Google Robot, while performing similarly on the other tasks. Its main failure reason is similar to Hyper-VLA due to inaccurate grasping. However, it also makes some other obvious mistakes, such as the robot arm getting stuck in the air, and not picking the target object up as expected after successfully grasping it. We also find that the robot arm movement controlled by OpenVLA is less smooth than HyperVLA, possibly due to its autoregressive way of predicting discretized action tokens.

For the other two weaker baselines RT-1-X and Octo, in addition to the grasping error, they sometimes even can not correctly locate the object to manipulate, or misunderstand the language instruction semantically, such as moving a wrong object to a wrong target object in the moving tasks, possibly due to that they are not built upon language and vision foundation models with strong generalization ability.

