# OpenReview forum: "HyperVLA: Efficient Inference in Vision-Language-Action Models via Hypernetworks"
_ICLR.cc/2026/Conference — Submitted to ICLR 2026_

### Official Review · Reviewer_VKKj · 2025-10-28

**Soundness:** 2
**Presentation:** 3
**Contribution:** 2
**Rating:** 4
**Confidence:** 4

**Summary:**

This paper presents HyperVLA, a vision-language-action (VLA) framework that leverages a hypernetwork to generate lightweight, task-specific policies for efficient inference.

**Strengths:**

- A clear objective—retaining VLA-level generalization while greatly improving inference efficiency.
- A concrete HN-based realization with sensible design choices (vision backbone, context-embedding normalization, linear action head) and strong speedups on SIMPLER/LIBERO with competitive success rates.

**Weaknesses:**

- Context dependence & robustness. The HN conditions on only the initial image and language, and the base policy omits language input. Please evaluate robustness to distribution changes like viewpoint/lighting shifts. *I strongly recommend including real-world experiments to further demonstrate the applicability and reliability of the proposed approach.*
- Accounting for HN cost. Table 3 excludes the one-time HN/encoder computation. Please report amortized wall-clock when tasks switch frequently vs. rarely, and how speedup scales with episode length.
- Limitations. A dedicated limitations section would help (e.g., potential brittleness to context errors, limited expressivity of a ~0.1M-parameter base policy for highly multimodal action distributions)
- Only partial setting on the simplerenv benchmark are presented. Please clarify the reason for not reporting the full results.

**Questions:**

1.	Hypernetworks have rarely been effective as an acceleration mechanism in other AI domains. Could the authors elaborate on what makes the VLA setting different—what specific properties of vision-language-action control allow this approach to succeed where it typically does not? A brief discussion of these distinctions and relevant historical context would strengthen the paper’s motivation.
2.	Regarding the question raised in the paper “Can we learn a generalist policy that combines the strong generalization ability of VLAs and the efficient inference of single-task policies?”. Are there any other possible technical routes besides hypernetworks?
For instance, can DeeR-VLA be viewed as an alternative realization of this idea? How do dynamic-routing approaches like DeeR-VLA compare with hypernetworks in terms of their advantages and disadvantages?

---

### Official Review · Reviewer_RdTx · 2025-10-29

**Soundness:** 3
**Presentation:** 3
**Contribution:** 3
**Rating:** 6
**Confidence:** 2

**Summary:**

This paper introduces HyperVLA, an efficient VLA model that leverages a Hypernetwork (HN) to dynamically generate lightweight task-specific policy networks. The design reduces inference latency from hundreds of milliseconds to ~4 ms while maintaining performance comparable to OpenVLA on SIMPLER and LIBERO, with well-organized ablations supporting its effectiveness.

**Strengths:**

- The work proposes a new Hypernetwork-based VLA design that achieves remarkable inference efficiency while preserving zero-shot generalization across diverse manipulation tasks.
- The work provides well-designed algorithmic components—pretrained vision backbone, HN normalization, and a lightweight action generation strategy—each validated through ablations to be essential for stable and efficient training.

**Weaknesses:**

- Lack of real-world validation: All evaluations are conducted in simulation (SIMPLER, LIBERO). Given the sensitivity of HN to distribution shifts, this omission limits confidence in real-world robustness and transferability to physical robots.
- Incomplete efficiency comparison: It is encouraged to include comparisons with recent optimized or lightweight variants of existing models that also improve inference efficiency. For example, OpenVLA-OFT demonstrates strong performance on the LIBERO benchmark while achieving notable gains in computational efficiency.
- Unexplored efficiency–capacity trade-off: The paper lacks analysis of how HN or policy capacity affects performance and scalability. It remains unclear how changing the HN hidden size impacts stability and generalization, whether the 0.1M-parameter base policy is a bottleneck, and how the method scales with larger datasets or more complex task distributions.

**Questions:**

- How consistent are the policies generated by the HN across tasks with similar instructions or visual contexts? For instance, would a policy generated for grasping an apple still succeed when applied to grasping an orange or a cup?
- The method generates compact task-specific policies without language conditioning at inference. Can this design handle complex or compositional instructions—such as spatial reasoning tasks (e.g., “place the left apple into the right basket”)?

---

### Official Review · Reviewer_6UDx · 2025-11-01

**Soundness:** 3
**Presentation:** 2
**Contribution:** 3
**Rating:** 6
**Confidence:** 4

**Summary:**

This paper addresses the challenge of high inference costs in existing VLA models. The authors propose a novel hypernetwork-based architecture that activates only a small, task-specific policy during inference. Several key algorithmic designs are introduced to enable the successful training of hypernetwork-based models, including leveraging prior knowledge from vision foundation models, applying hypernetwork normalization, and designing an action generation strategy. The proposed approach is evaluated in simulation environments such as SIMPER and Libero, demonstrating its effectiveness.

**Strengths:**

1. The paper introduces a novel hypernetwork-based VLA model structure that activates only a compact, generated policy at inference, significantly improving efficiency.

2. To address the optimization challenges of hypernetworks, the authors analyze the differences between HN training and standard training, propose normalizing the context embedding, and validate the effectiveness of this strategy through experiments.

3. The paper further highlights practical techniques for training HN-based VLAs, such as utilizing prior knowledge from vision encoders and employing a lightweight linear action head to improve efficiency.

4. The experiments are well-organized and convincingly demonstrate the proposed model’s generalization, adaptability, and inference efficiency compared with mainstream VLAs such as RT1-X, Octo, and OpenVLA.

**Weaknesses:**

1. Hypernetworks for policy generation are not entirely new, with related works in meta-RL and dual-system VLAs sharing conceptual similarities. The paper could clarify more explicitly how HyperVLA advances beyond these prior approaches.

2. The role of the context token as an input to the context encoder seems important, but the current explanation could be elaborated further for clarity.

3. The linear action head combined with action chunking appears to contribute significantly to inference efficiency. However, a more detailed analysis or comparison highlighting its advantages would further strengthen the contribution.

4. The evaluation focuses on simulation environments (SIMPER and Libero). While this provides valuable insights, demonstrating the method’s inference efficiency on real robotic platforms could make the results more compelling.

**Questions:**

1. How does HyperVLA perform in real-robot settings with strict latency constraints? Is the 120× speedup sufficient for high-frequency control (e.g., >50 Hz) in practice?

2. How sensitive is HyperVLA to the choice of backbone? For example, what happens if weaker vision encoders (e.g., EfficientNet) are used instead of DINOv2?

---

### Official Review · Reviewer_3cYT · 2025-11-02

**Soundness:** 2
**Presentation:** 3
**Contribution:** 1
**Rating:** 2
**Confidence:** 4

**Summary:**

This paper proposes HyperVLA, an approach to make VLA models more efficient at inference time by introducing a hypernetwork that dynamically generates policy weights conditioned on task inputs. The goal is to reduce inference latency and computational cost without significantly hurting performance. The authors evaluate HyperVLA on simulation benchmarks like SIMPLER and LIBERO, reporting large speedups and comparable success rates to full VLA baselines. The work aims to make VLAs more practical for real-world robotic control by improving efficiency and deployability.

**Strengths:**

1. The paper tackles an important and timely problem . VLA models are getting very capable but are often too heavy for real-time control. The motivation to improve inference efficiency is well grounded and practically relevant.
2. Using a hypernetwork to generate task-specific weights for the policy is a neat idea. Even if not entirely novel, it’s an intuitive way to adapt to diverse tasks without keeping multiple large models in memory.
3. The paper reads clearly and the technical sections are well-structured. The figures and tables are also clean and informative.

**Weaknesses:**

1. The main idea of using a hypernetwork to make inference faster  is interesting, but it’s not clear why this specific route was chosen over more standard acceleration methods. Approaches like consistency distillation [Consistency Policy], flow matching action head [Pi series or GR000T series], and quantization-based compression are well established and might offer simpler alternatives. The paper doesn’t really compare against them or argue theoretically for why HNs are better suited. As a result, the design choice feels more empirical than principled.
2. The paper mostly compares against RT-1-X, Octo, and OpenVLA. They are solid but somewhat dated baselines. Recent models like Pi0, MiniVLA, or DeeR-VLA leverage large-scale multimodal pretraining and show strong scaling behavior, which is one of the core advantages of current VLAs. Not including these makes it hard to judge how HyperVLA performs in the context of modern scaling trends. The explanation that those models use different datasets isn’t really sufficient.
3. Everything is evaluated in simulation (SIMPLER, LIBERO), and there’s no hardware validation. For a paper that positions itself as making VLAs practical for robotics, that’s a big gap. Real-world deployment involves latency, sensing noise, and control delays that simulation doesn’t capture. Even a small-scale physical experiment, like what Octo did, would make the claims much stronger.
4. Most of the architectural decisions: freezing the vision backbone, using a linear MLP head and training with MSE, are standard tricks from prior work like Llava and OpenVLA. These choices seem empirical rather than offering new insight. The method works, but it feels more like careful system tuning than a genuinely new idea.

**Questions:**

1. Have you compared HyperVLA’s trade-offs against methods like distillation or quantization, especially in terms of accuracy vs. compute?
2. The current baselines are weak. How would the method behave when scaling up with hundreds of millions of multimodal samples? Does the hypernetwork remain efficient?
3. Any evidence on real-robot latency and performance?

---

### Official Review · Reviewer_1L2x · 2025-11-09

**Soundness:** 2
**Presentation:** 2
**Contribution:** 3
**Rating:** 2
**Confidence:** 4

**Summary:**

The authors introduce HyperVLA, a hypernetwork (HN)-based architecture that activates a small, task-specific policy during inference, which significantly reduces inference costs.

**Strengths:**

•The paper is well-written with a clear and logical structure.
•The approach demonstrates a notable improvement in inference speed, effectively achieving a lightweight version of VLA.

**Weaknesses:**

•Baseline Selection and Fairness:
￮The choice of baselines is questionable. The paper would benefit from comparing against more recent and stronger models, such as OpenVLA-OFT, pi0, pi0-FAST, and pi05-kI. These models, to the best of my knowledge, achieve higher performance on the LIBERO benchmark, which makes it difficult to demonstrate the advantages of the proposed approach.
•Inference Cost Considerations:
In Section 4.4, line 400, the authors claim that the computational cost of HN is negligible since they are only activated once at the beginning of each episode.  However, this assessment may overlook the potential impact of these costs in different scenarios, particularly given that the tasks evaluated are short (e.g., pick-place tasks).
It would be helpful if the authors could:
1. Provide the results of the HN generation cost.
2.Offer a more detailed analysis and discussion on how the model performs in longer tasks with task-switching.
•Real-World and Complex Task Evaluation:
￮The authors should include results on real-world experiments and more complex, longer tasks to better demonstrate the advantages of inference acceleration in practical settings.

**Questions:**

•Have the authors considered alternative methods like soft prompting, LoRA, or MoE (Mixture of Experts) to sparsely activate VLA parameters, which are common in recent approaches?
•Could the authors comment on the statement in lines 156-158, where it is mentioned that existing VLA models require "activating the whole model during both training and inference"? Is this claim still valid, given recent developments in VLAs that decouple tasks more efficiently?

---

### Meta-Review · Area_Chair_7iHc · 2025-12-20

**Summary:**

This paper received 2,2,4,6.  The authors did not provide a rebuttal.  Key concerns include weak experiments, unclear approach design choices, and novelty concerns.  Since there was no rebuttal, the AC sees no reason to recommend acceptance.

**Reviewer Scores:**

N/A

---

### Decision · Program_Chairs · 2026-01-26

Reject